# Morphological and Genetic Assessment of Invasive *Corbicula* Lineages in Southern South America: A Case Study in Argentina

**DOI:** 10.3390/ani14131843

**Published:** 2024-06-21

**Authors:** Leandro A. Hünicken, Esteban M. Paolucci, Pablo D. Lavinia, Francisco Sylvester

**Affiliations:** 1Consejo Nacional de Investigaciones Científicas y Técnicas (CONICET), Buenos Aires C1425FQB, Argentina; estebanmpaolucci@gmail.com (E.M.P.); pablo.lavinia@conicet.gov.ar (P.D.L.); franciscosylvester@gmail.com (F.S.); 2Museo Argentino de Ciencias Naturales ‘Bernardino Rivadavia’, Av. Ángel Gallardo 470, Buenos Aires C1405DJR, Argentina; 3Instituto Para el Estudio de la Biodiversidad de Invertebrados (IEBI), Facultad de Ciencias Naturales, Universidad Nacional de Salta, Av. Bolivia 5150, Salta A4408FVY, Argentina; 4Laboratorio de Investigación y Conservación de la Biodiversidad (UNRN-InCoBIO), Universidad Nacional de Río Negro, Sede Atlántica, RPNº1 y Rotonda de la Cooperación, Viedma R8500JCG, Argentina; 5CIT Río Negro (UNRN-CONICET), Universidad Nacional de Río Negro, Sede Atlántica, Viedma R8500JCG, Argentina; 6Faculty of Biological Sciences, Goethe University Frankfurt, Max-von-Laue Str. 13, 60438 Frankfurt am Main, Germany

**Keywords:** *Corbicula*, morphology, COI, Argentina, distribution

## Abstract

**Simple Summary:**

The global distribution of the genus *Corbicula* is driven by multiple hermaphroditic lineages. In Argentina, there remains a lack of comprehensive understanding of the current distribution, identity, and genetic relatedness of invasive *Corbicula* lineages. Reviewing 15 populations, we discriminate extant lineages based on both morphology and genetics in Argentina. We identified two lineages: AR morphotype (FW5 haplotype) and CS morphotype (FW17 haplotype). These lineages exhibited virtually segregated distributions, however, intermediate morphotypes found in northeastern Argentina suggest the presence of hybrids. Our study clarifies *Corbicula* identity and distribution, offering insights into invasion patterns in a wide geographic region.

**Abstract:**

The broad global distribution of freshwater clams belonging to the genus *Corbicula* is driven by multiple hermaphroditic lineages. These lineages, characterized by shared morphological traits and phenotypic plasticity, pose challenges to morphological identification. Genetic markers, such as the mitochondrial COI gene, play a crucial role in delineating these lineages and their ranges. Morphotypes represent observed phenotypic variations, while lineages are defined based on genetic markers. Here, we comprehensively review *Corbicula*’s distribution in Argentina, discriminate extant lineages based on both morphological and genetic (COI) data, and describe variations in internal and external morphologies using 15 Argentine populations. Genetic analyses identified two mitochondrial lineages: the AR morphotype (FW5 haplotype) and CS morphotype (FW17 haplotype). Strikingly, despite having similar vectors, origins, and invasive stages, *Corbicula* lineages exhibit virtually segregated distributions. However, mitochondrial haplotypes are found in sympatry mainly in northeastern Argentina where individuals with intermediate morphotypes exist, suggesting the presence of hybrids due to maternal genome retention. These findings contribute to the clarification of the identity and distribution of *Corbicula* lineages in Argentina, where the genus has been found for over half a century. Similar studies are needed in other areas to better understand the invasion patterns of this successful and adaptable group.

## 1. Introduction

The freshwater bivalve genus *Corbicula*, first described by Megerle von Mühlfeld in 1811, is native to Africa, Asia, Australia, and the Middle East [1]. However, the fossil record indicates the presence of *Corbicula* in North America during the Eocene epoch, as well as multiple incursions into freshwater ecosystems in northwest Europe during the Pleistocene interglacial periods, demonstrating the group’s significant dispersal capacity [2,3]. Consequently, it is not surprising that given global interconnectedness and species movement across distant regions, *Corbicula* populations are now established in freshwater ecosystems throughout North and South America and Europe [4]. A further worldwide expansion of suitable habitats for invasive clams of the *Corbicula* genus, such as *Corbicula fluminea* (Müller 1774), has been forecasted in the context of climate change [5].

While in the native range, sexual and asexual species coexist, and the recent proliferation of certain lineages has been facilitated by androgenesis, a particular reproductive strategy involving clonal asexual reproduction [6]. The widespread geographical and environmental distributions of these androgenetic lineages correlate with significant morphological diversity, particularly in taxonomic traits such as shell shape and color [7], leading to taxonomic complexity and confusion with numerous species being described primarily based on shell characteristics [8]. As a result, the prevailing scientific consensus opts to use the term “forms” to denote the various morphological variations observed in *Corbicula*. In Europe, three forms are recognized: R, Rlc, and S [9,10], which correspond to morphotypes A, B, and C, respectively, described in the Americas [11,12]. Recent molecular and genetic studies of the mitochondrial cytochrome c oxidase subunit I (COI) gene have shed light on the phylogeny of these forms in both their invasive and native ranges. These studies indicate that the AR Form (commonly identified as *C. fluminea*) aligns with the native FW5 haplotype, prevalent in East Asia (Japan, Korea, Taiwan, China, Indonesia, and Myanmar [13]. The CS lineage is homologous to the FW17 haplotype [13], and its origin is cryptic but suggested in African populations [14]. There is a distinction within the RlcB form as well; while the North American form B corresponds to the native FW1, the European Rlc form is related to the FW4 haplotype, both of which are present in Korea [13]. Moreover, alterations in the external morphology in bivalves have been linked to internal variation in gill and palp areas [15,16]. Despite the critical roles these organs play in the reproduction and physiology of *Corbicula* species [17,18], their association with inter-specific differences or phenotypic plasticity remains largely unexplored.

Genetic characterization of these lineages does not fully resolve their taxonomy, partly because the same haplotype may occur in different morphotypes. This ambiguity could stem from phenotypic plasticity or androgenesis [12]. In the latter, hermaphroditic individuals produce biflagellate diploid sperm that, during fertilization, expel the maternal nucleus, expressing solely the paternal genotype but retaining maternal mitochondrial DNA [19]. Furthermore, sperm from one lineage can fertilize oocytes from another, leading to “egg parasitism” and cytonuclear mismatches [12,20]. In some cases, this results in offspring with a hybrid genome due to incomplete extrusion of maternal nuclear DNA [21], resulting in triploid progeny with both maternal and paternal nuclear DNA. Hence, reliance solely on mitochondrial DNA analysis may introduce bias, and an integrated taxonomic approach, incorporating examination of morphological and anatomical data, is recommended [22].

Recent research has comprehensively identified the invasive lineages of *Corbicula* spp. in South America using both morphological and genetic traits [23] although the geographical coverage remains uneven across the subcontinent. In Argentina, genetic studies have been based on the mitochondrial gene COI [23] and the nuclear ribosomal gene 28S [12], limited to certain individuals from the Río de la Plata, upper Río Negro Basin, and Iguazú Falls. This constitutes a significant gap given the size of the country and the widespread distribution of *Corbicula*. in it. The populations analyzed, morphologically assigned to *C. largillierti* (CS lineage), *C. fluminea* (AR lineage), and *Corbicula* sp., exhibited the mitochondrial COI haplotypes FW17, FW5, and FW1, respectively [12,23]. Notably, the latter mitochondrial haplotype (commonly found in individuals of form B) was detected in clams with nuclear haplotype C, in hybrid specimens between B and C, and was morphologically assigned to the CS form. The extensive distribution observed in Argentina might correspond not to one but a range of different mitochondrial lineages that remain yet to be characterized. Additionally, the presence of hybrids as a result of maternal nuclear capture, and morphogenetic inconsistencies due to cytonuclear mismatches, underscores the likely complexity of combinations and the need for molecular tools to accurately identify populations.

Despite various ecological, taxonomic, and genetic investigations there remains, however, a lack of comprehensive understanding of the current distribution, identity, and genetic relatedness of invasive *Corbicula* lineages in Argentina. In this study, we systematically analyzed the distribution of *Corbicula* spp. across a broad region of South America through a comprehensive examination of collections, literature, and our own field sampling. We used molecular markers to identify the populations and scrutinize their morphology. The principal objectives of this study were to (1) update the distribution, (2) delineate the morphological and anatomical characteristics, (3) evaluate the genetic diversity and phylogeography of *Corbicula* spp. using mitochondrial DNA, and (4) assess the correlation between morphological variation and molecular identity of the genus in Argentina.

## 2. Materials and Methods

### 2.1. Distribution of Corbicula in Argentina: Review and Sampling

The distribution of *Corbicula* in Argentina was compiled through an extensive review of malacological collections, literature sources, and expert consultations [24,25,26,27,28,29,30,31,32]. The primary collections consulted included the Museo de La Plata (MLP), the Museo Argentino de Ciencias Naturales “Bernardino Rivadavia” (MACN), and the Fundación Miguel Lillo (FML). To gather current distribution data for *Corbicula* and collect specimens for subsequent analyses, a nationwide sampling initiative was conducted, covering the estuaries of the Negro and Río de La Plata rivers, the central and northwest basins, and the main plain rivers of the La Plata Basin from 2015 to 2017 (Table 1). The collected specimens were deposited in the MACN Invertebrate Collection for further analysis.

### 2.2. Morphological Assignment of Lineages

Despite the difficulties previously mentioned, the assignment of *Corbicula* morphotypes can be carried out when certain characteristics remain relatively stable and, especially when extreme states are observed. The distinction between the two previously reported forms in Argentina, CS and AR, relies on factors such as internal and external coloration, depth and number of ribs, the shape of the posterior margin and umbo, and thickness of the valves [29,33,34] (Figure 1). A conservative approach was adopted, whereby if any character exhibited ambiguities or contradicted others, it was not definitively assigned to either of the two forms and was designated as an “intermediate” form (form I).

### 2.3. Shell Morphology

Shell length (L), width (W), and height (H) were measured for each individual using a caliper (precision: 0.01 mm) to estimate the maximum anteroposterior, lateral, and dorsoventral dimensions, respectively. Shell weight (SW) was measured after the shells were dried at room temperature in a desiccator following Coughlan et al. [35]. The valves were scanned to capture their internal and external surfaces at the highest possible resolution (2400 dpi). Subsequently, the number of external concentric ribs per centimeter (Ribs, cm⁻^1^) was quantified from the obtained images. Additionally, to incorporate the internal and external colors of the valves as quantitative variables, RGB values were extracted through color queries conducted using Adobe^®^ Photoshop^®^ (2017). This process involved sampling three random sectors of the internal and external faces of the valves, each with a sample area of 101 pixels.

### 2.4. Internal Morphology

Gills and palps of each clam were carefully dissected and placed in a Petri dish with ethanol, before being photographed using a dissecting microscope (Leica S8APO; 10× magnification; Figure 2). The gill and palps areas were quantified by digitally outlining their contours using Digimizer version 5.3.5 (Ostend: MedCalc Software 2016; precision: 0.1 mm^2^) [36]. For the gills, both the internal and external demibranchs of one randomly chosen gill (left or right) were measured, and their combined areas were doubled to determine the total gill area (GA). Regarding the palps, all four were individually measured, including external and internal structures on both sides. Due to their curved shape, the palps were oriented with the concave side facing up, and their edges were extended to ensure they lay flat. Filament density (FD), defined as the average number of filaments per millimeter, was assessed in each clam (30 per population). Three randomly selected sections from a 50× magnification image of the gills (Figure 2D) were used to calculate FD, following the methodology outlined by Paolucci et al. [16].

### 2.5. Population Selection for Molecular Analysis: Geographic Diversity across Argentina

Genetic analyses were conducted utilizing samples obtained from the previously mentioned collecting trips. We selected individuals from 15 populations to ensure broad geographic representation, covering diverse forms of *Corbicula* in Argentina. These populations are drawn from five regions (refer to Table 1 for site codes and locations): the Río de la Plata estuary (PLA, BUE); Northeast (CON, APO, ASJ, SRO, ESP, RML); Central (COR); Northwest (DCC, MOL, RJU, SDE, CAT); and Patagonia (RNE).

### 2.6. DNA Extraction, PCR Amplification, and Sequencing

For a total of 225 individuals (15 specimens per sampling location), genomic DNA was obtained following a glass fiber-based extraction protocol [37,38,39]. A 658 bp fragment near the 5′ end of COI was amplified using BivF4_t1 (5′-TGTAAAACGACGGCCAGTGKTCWACWAATCATAARGATATTGG-3′) and BivR4_t1 (5′-CAGGAAACAGCTATGACTAMACCTCWGGRTGVCCRAARAACCA-3′) pair of primers [40]. Thermocycling conditions consisted of 2 min at 94 °C for 2 min; 5 cycles of 30 s at 94 °C, 40 s at 45 °C and 1 min at 72 °C; 35 cycles of 30 s at 94 °C, 40 s at 51 °C and 1 min at 72 °C for; final extension of 10 min at 72 °C. Sequencing was performed bidirectionally at the Centre for Biodiversity Genomics (Guelph, ON, Canada) for 36 individuals, and unidirectionally for 190 individuals at Macrogen (Seoul, Republic of Korea), with the same primers used for amplification. Sequences were edited using CODONCODE ALIGNER (CodonCode Corporation, Centerville, MA, USA). We obtained good-quality sequences for 211 individuals, which can be found in GenBank under the accession numbers (PP827592–PP827802; Appendix A).

### 2.7. Data Analyses

#### 2.7.1. Phylogenetic and Phylogeographic Analyses

To increase our taxonomic and geographic coverage for the phylogenetic analyses, we mined from GenBank 58 COI public sequences representative of other native and invasive *Corbicula* species and mitochondrial lineages or haplogroups (see Appendix A—Appendix A). A gene tree based on 269 sequences was inferred through Bayesian methodology using MrBayes 3.2.2 [41]. The best-fit model nucleotide evolution for our COI data set was TPM3uf+G based on the Bayesian information criterion (BIC) as implemented in jModelTest 2.1.1 [42]. We specified in MrBayes the closest model available (HKY+G) and conducted two independent runs of 10 million generations under default priors and sampling trees every 100 generations. The average standard deviation of split frequencies between runs was <0.01, indicating convergence. We verified that both runs reached stationarity and that we had a good sample of the posterior probability distribution with Tracer 1.7.2 [43]. We discarded the first 25% of sampled trees as burn-in and combined the remaining 75,000 topologies of each run to generate a 50% majority rule consensus tree. The tree was rooted with *Neocorbicula limosa*.

To further investigate the two major freshwater (FW) haplogroups found in Argentina, we generate an unrooted statistical parsimony network with TCS 1.21 [44] and a cut-off value of 95% [45]. We also include some specimens of *C. sandai* and other FW lineages for context. The obtained network was edited with tcsBU [46]. We used MEGA [47] to estimate the mean uncorrected genetic distance (p-distance) between haplogroups, and the per-site nucleotide diversity (π) within haplogroups in DNASP 5.10 [48]. In both cases, we used the pairwise deletion option for missing data. Finally, we explored the correspondence between mitochondrial and morphological identities and the geographic distribution of the different morphotypes and haplogroups found in Argentina. To do so, we first assessed how representatives of each of the three phenotypic forms (AR, CS, and I) are clustered within the two mitochondrial lineages, and then, how both lineages and morphotypes are distributed across sampling locations.

#### 2.7.2. Morphological Clustering Analysis and Morphotype Reassignment

We assessed morphological variations among *Corbicula* populations through multivariate clustering analysis, standardizing morphological variables by size. Specifically, linear measurements of the valves were expressed as ratios between Height and Length (H:L), as well as Width and Length (W:L). Furthermore, we adjusted shell weight by dividing it by the general size, calculated as the product of the length of the three major body axes (L × H × W), a method known to reliably estimate shell thickness (ST, [17]). Gill area and palp area were also normalized relative to the general shell size of the valves (RGA and RPA, respectively). Additionally, the palp-to-gill ratio (PA:GA) was included in the analyses. Before incorporating color information, principal component analysis (PCA) was performed on the six color variables obtained previously (red, green, and blue composition for both internal and external shell faces), with retention of the first two principal axes (PC1_color and PC2_color). The scores of each clam in these axes were then incorporated as variables. Finally, we included the number of concentric ribs and filament density to create a matrix of ten morphometric variables.

We utilized fuzzy clustering to partition the data into clusters. Fuzzy clustering is a method employed to group data points into clusters, where membership in each cluster is not strictly binary, allowing for partial membership of data points in multiple clusters [49]. Unlike traditional “hard” clustering algorithms such as K-means, which assign each data point exclusively to a single cluster, fuzzy clustering enables more flexible assignments. We opted for this approach due to the potential presence of intermediate morphotypes within the *Corbicula* genus, attributed to cytonuclear inconsistencies and hybridization between lineages resulting from the androgenetic reproduction method. In fuzzy clustering, each data point is assigned a membership value for each cluster, indicating the degree of association with that cluster. These membership values typically range between 0 and 1, with higher values indicating stronger membership. The analysis was made by the *fanny* function from the ‘cluster’ package 2023 [50]. Before this, we determined the optimal number of clusters by the within-cluster sums of squares method, employing the *fviz_nbclust()* function from the ‘factoextra’ package v. 1.0.7 2020 [51]. The analyses consistently indicated two as the most probable number of clusters, coinciding with the recognized forms observed in the study area, therefore the number of clusters for fuzzy clustering was set to two. We obtained the membership probabilities for each cluster and established a cut-off threshold of 0.65 to determine membership in either cluster. Individuals failing to exceed this limit were classified as not belonging to any specific cluster and categorized as “intermediate”. These results were compared with the previously established assignments to assess the classification’s effectiveness and correspondence between both methods.

Lastly, we explored the underlying structure of the dataset and identified patterns among the variables that best captured the variability between forms and populations conducting a principal component analysis (PCA). We identified the most relevant morphometric variables for distinguishing *Corbicula* morphotypes and examined the positioning of clusters derived from the preceding analysis, along with that of intermediate individuals.

## 3. Results

### 3.1. Distribution of Corbicula in Argentina: Review and Sampling

From collections and bibliography, 157 Corbicula records in Argentina were compiled, predominantly belonging to the AR form (*C. fluminea*, 93), with the remainder comprising CS form (*C. largillierti*, 49), a hybrid form (4), and unidentified records (11) (Appendix A Appendix A). Based on the data in hand, the temporal progression seems to indicate that between the 1980s and 1995, both Corbicula lineages rapidly spread northward along the Paraná River and lower Uruguay, primarily recorded in Santa Fe, Entre Ríos, Chaco, and Corrientes provinces. Subsequently, they underwent sudden geographical expansion into diverse regions, occupying basins unconnected to their original habitats (Appendix A Appendix A). Finally, lineages exhibited distinct geographical distributions, with the AR form concentrated in the east and southern main plain rivers, while the CS form was found mainly in reservoirs and low-order rivers in the Central and Northwestern regions. Except for a few coexisting populations, lineages primarily inhabited allopatric areas. Notably, the Northeast featured populations with intermediate morphotypes (Appendix A Appendix A). Our own surveys encompassed 26 populations, including previously unreported sites, with 11 corresponding to the AR form, 8 to CS, and 7 to the intermediate form (Appendix A Appendix A).

### 3.2. Phylogenetic and Phylogeographic Analyses

The Bayesian gene tree recovered the estuarine and freshwater (FW) Corbicula clades as reciprocally monophyletic with maximum node support (Figure 3 and Appendix A). All individuals sampled in Argentina for this study clustered within two of the four major freshwater (FW) mitochondrial lineages of the androgenetic, invasive Corbicula clams: FW5 and FW17 (Figure 3). Specimens were almost equally distributed between the two haplogroups, being FW5 slightly more frequent than FW17 in Argentina (Table 2). Both lineages were recovered as monophyletic with high (0.99) to maximum (1.0) support. The new, recently reported FWBRA lineage from Brazil [23], was recovered as an intermediate haplotype between FW5 and FW17, being slightly closer to the former in terms of genetic divergence (Figure 3). The mean, uncorrected genetic distance between FWBRA and FW5 was 1.25%, and it was 1.57% between the former and FW17.

Individuals within the FW5 haplogroup corresponded to the phenotypic form AR, while individuals categorized as CS clustered within the FW17 lineage (Figure 3, Figure 4, and Appendix A; Table 2). The only exceptions were three specimens with a phenotype AR from Buenos Aires province that were grouped within the FW17, and three representatives of the CS from Cordoba Province that clustered within the FW5 haplogroup (Figure 4 and Appendix A). As for the specimens with intermediate forms, 53 individuals clustered inside the FW5 lineage and 30 within the FW17 haplogroup (Figure 4 and Appendix A, Table 2). Representatives of the form CS were concentrated in northwestern Argentina, while specimens with an AR phenotype were distributed to the east and south, and individuals with intermediate phenotype were found almost exclusively in northeastern Argentina. No sampling location held more than one phenotypic form (Figure 4A).

The haplogroup FW5 was found almost exclusively in eastern Argentina, from Misiones province in the north to northern Patagonia (Río Negro province) in the south (Figure 4A). In contrast, the FW17 lineage was chiefly distributed in central and northwestern Argentina. Haplogroups were found in sympatry in five sampling locations (Figure 4A), one in central Argentina (COR) and the remaining four to the east of the country (ASJ, ESP, PLA, RML). Finally, and as evidenced by the haplotype network, genetic diversity within both lineages was remarkably low, with a common central haplotype and a few, unique haplotypes closely related (one or two mutational steps) to it (Figure 4B; Table 2).

### 3.3. Morphological Clustering Analysis and Morphotype Reassignment

The cluster analysis exhibited strong agreement with our initial assignments, with 82.5% (245 out of 297) of individuals classified consistently into either the AR, CS forms or intermediate categories. Notably, discrepancies primarily emerged in Northeastern populations (Misiones, Formosa, and Chaco), where 32 individuals initially classified as intermediate was identified with a greater than 65% probability of belonging to the AR cluster. Similarly, at the DCC site, 7 out of 30 initially designated as CS forms are now categorized as intermediate.

The first three principal components (PCs) of the PCA collectively explained 71.19% (PC1 = 34.18%, PC2 = 19.69%, and PC3 = 17.32%) of the variance in the dataset, effectively summarizing its variability (Table 3). The PCA revealed distinct patterns in the morphometric variables of Corbicula populations: PC1 exhibited strong positive loadings for internal color and number of ribs, and strong negative loadings for ST and W:L ratio, distinguishing clusters associated with AR and CS forms (Figure 5). PC2 demonstrated significant positive loadings for RGA and negative loadings for H:L ratios, highlighting differences between populations. PC3 showed a notable correlation with the external color of valves, with intermediate individuals characterized by positive values (Figure 5).

## 4. Discussion

This study presents the first review of *Corbicula* distribution in Argentina, encompassing a significant subcontinental region. Previous research on *Corbicula* in Argentina, using morphological and genetic data, has been limited to a single basin or a few specific environments [12,23,29,52]. Our review coupled an unprecedented reevaluation of taxonomic identification within malacological collections with a countrywide sampling effort for morphological and molecular analyses. Our findings reveal a prevalence of the AR form, particularly in the east and southern main plain rivers, contrasting with the dominance of the CS form mainly in reservoirs and low-order rivers in the central and northwestern regions. Surprisingly, despite similarities in vectors, geographical origins, and the temporal stages of invasion, distinct *Corbicula* mitochondrial lineages exhibit virtually segregated distributions in Argentina. This divergence likely indicates other underlying factors such as morphological, physiological, or ecological disparities. Notably, the northeast region displayed populations with intermediate morphotypes, suggesting potential hybridization or transitional characteristics.

Prior to our current study, mitochondrial identities had been established for three Argentinean populations: the FW5 and FW17 lineages coexisting in the Río de la Plata estuary, the FW17 lineage in the upper Río Negro basin in Northern Patagonia, and the FW1 lineage at the Iguazú Falls on the Argentine–Brazilian border [12,23]. Our phylogenetic analyses, covering 15 populations across the entire distribution range of *Corbicula* in Argentina, highlight the widespread prevalence of two freshwater mitochondrial lineages, FW5 and FW17. Most populations (10 out of 15) consist of a single mitochondrial lineage, revealing a clear pattern of geographic segregation among them. This discovery is particularly striking given the typical sympatric distribution of both lineages across Europe, North America [6,53], and Russia [54]. Furthermore, it is noteworthy that in Argentina, the FW17 lineage exhibits a remarkably widespread distribution, establishing abundant populations that have persisted for over two decades in regions not colonized by the AR form. This phenomenon is unique and contrasts with the lineage’s limited geographic range in Europe and North America [9,53,55,56]. Even in neighboring Brazil, the predominant lineage is FW5, with FW17 found in sympatry only in a few locations [23].

In general, Argentine lineages FW5 and FW17 exhibited a high correspondence with forms AR and CS, respectively, mirroring observations in other locations worldwide [6,12]. However, there are exceptions to this pattern. Instances of discordance between mitochondrial lineages and morphotypes have been documented in several *Corbicula* populations belonging to different forms [6,9,12,21]. These discrepancies suggest processes of androgenetic egg parasitism [6,12,21]. In the present study, cytonuclear mismatches were identified in Buenos Aires, where three individuals displaying an AR phenotype were classified under FW17, and in Cordoba, where three individuals with a CS phenotype clustered within the FW5 haplogroup. Currently, only one form is prevalent in each of these locations [52,57], historical records indicate the past occurrence of both forms [29,58]. Thus, the current presence of both mitochondrial haplotypes likely reflects a previous coexistence. A larger sampling effort than the one made here should be made to completely rule out the previous coexistence of forms in some of these locations. Another instance is illustrated by the intermediate morphotypes observed in northeastern Argentina. These morphotypes may represent hybrid individuals resulting from sympatric invasive lineages, possibly due to androgenesis with nuclear capture of the maternal DNA [9,14,21].

The morphological variations observed in this study are consistent with previous research, which primarily relied on the ratios of linear shell measurements, a characteristic and conserved trait relevant to morph differentiation [9,23,55,59]. In contrast, the relative gill area did not emerge as a significant variable in differentiating between morphs. This finding contrasts somewhat with previous research conducted in Argentina [17]. One reason for this disparity may be the different variables used to relativize the gill area. While [17] used dry weight, a physiologically meaningful variable, in our study, shell volume was utilized solely for the purpose of morphometric descriptors. Another contributing factor may lie in the scope of the studies: Hünicken et al.’s [17] research was based on only a few populations from two sites, potentially failing to capture some of the intra-morph variation. In the present study; however, this index explained substantial intracluster interpopulation variability along PC2 of the PCA. Notably, individuals unable to be assigned to either form spanned a broad range of morphometric space. This significant phenotypic variability could be attributed to the presence of various combinations of nuclear genes among lineages, resulting in diverse phenotypes. Nevertheless, alternative hypotheses such as phenotypic plasticity cannot be dismissed without further studies involving nuclear genes.

Phylogenetic and cytological studies have shown the absence of reproductive barriers between lineages, challenging the utility of the biological species concept in this group [19,22]. Consequently, the scientific consensus has leaned towards classifying *Corbicula* as a “complex of polymorphic species” [6]. However, this classification does not imply ecological equivalence among different forms; in fact, there are lineage disparities not only in morphology but also in microhabitat preferences [60,61], individual growth rates [62,63], reproductive event frequency, offspring release numbers [64], and physiological traits such as filtration [65] and metabolic rates [17]. The pronounced segregation of forms and haplotypes discovered across Argentina bolsters this notion. It has been suggested that some of the variability in life-history traits may have a genetic basis [11,61], although phenotypic plasticity could also be a contributing factor, as observed in other freshwater invasive bivalves [16]. Further research is necessary to determine their relative significance, as well as to assess the potential adaptive advantages conferred by these differences in various environmental conditions, and their implications for distribution patterns.

## 5. Conclusions

The current morphological and genetic assessment of *Corbicula* lineages across Argentina reveals a complex taxonomic and phylogenetic scenario surrounding this globally invasive genus. A key finding involves the distinct segregated distribution exhibited by *Corbicula* lineages that are often considered biologically and ecologically equivalent entities. The presence of morphologically intermediate forms that correspond to different genetic configurations adds further complexity to the situation. Additional molecular research using both mitochondrial and nuclear markers is necessary to clarify the complex phylogenetic picture involving numerous hybridization pathways. Moreover, physiological and ecological studies are essential to translate genetic differences into fitness characteristics, with implications for competitive ability and invasiveness.

## Figures and Tables

**Figure 1 animals-14-01843-f001:**
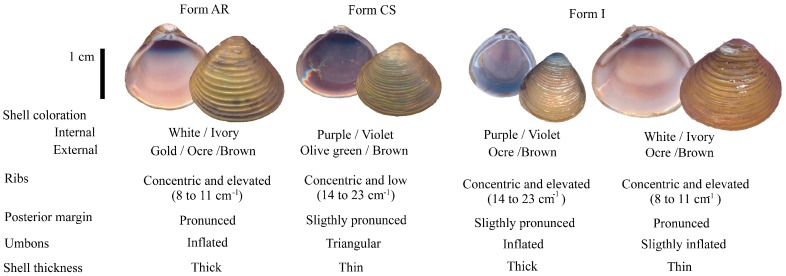
Internal and external views of the left valve of *Corbicula* form AR (*C. fluminea sensu lato*; Müller, 1774), form CS (*C. largillierti sensu lato*; Phillipi, 1844), and intermediate forms (Form I), along with the diagnostic characters utilized for morphological assignment of lineages in the present study.

**Figure 2 animals-14-01843-f002:**
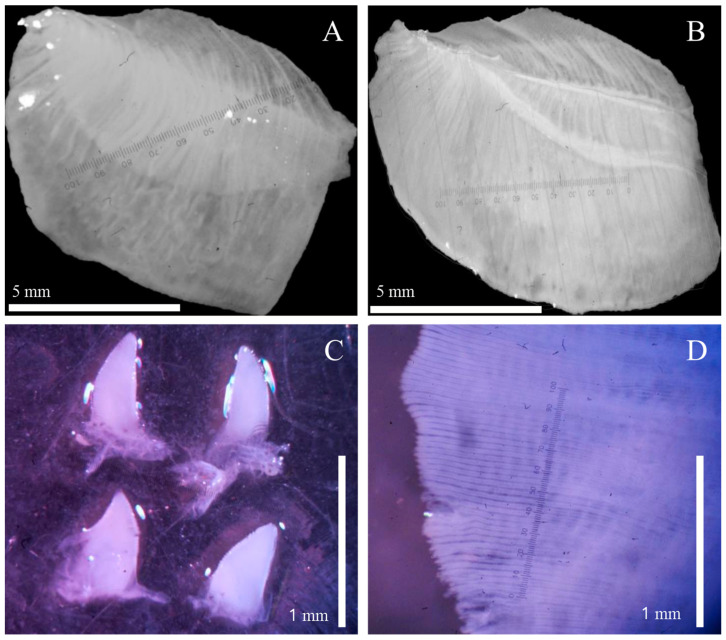
Examples of images taken to measure the gill area: (**A**) external demibranch, (**B**) internal demibranch), palp area (**C**), and filament density (**D**) in *Corbicula* specimens collected in Argentina between 2015 and 2017.

**Figure 3 animals-14-01843-f003:**
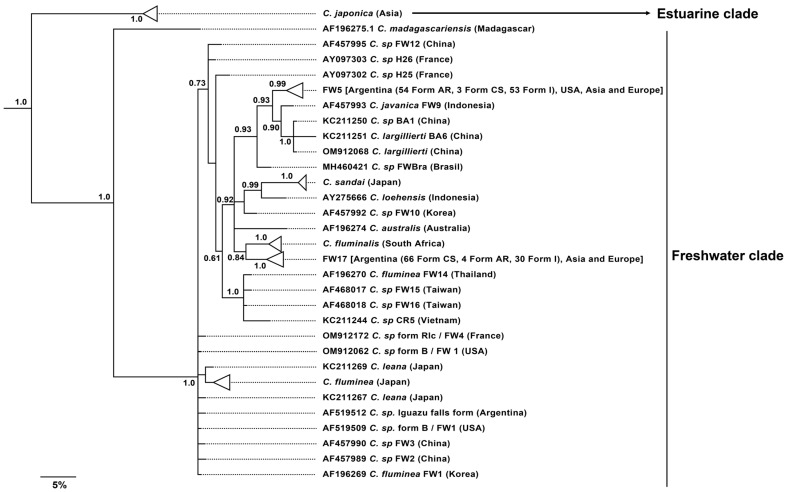
Bayesian majority rule consensus tree obtained from the analysis of 268 sequences of the mitochondrial gene COI (683 bp), showing the relationships among estuarine and freshwater species and major haplogroups of *Corbicula* clams. Numbers above or below the branches indicate Bayesian posterior probability node support. Some clades were collapsed and a few support values were omitted for clarity and simplicity (see Appendix A for the full topology with all tips shown). The tree was rooted with *Neocorbicula limosa* (Appendix A).

**Figure 4 animals-14-01843-f004:**
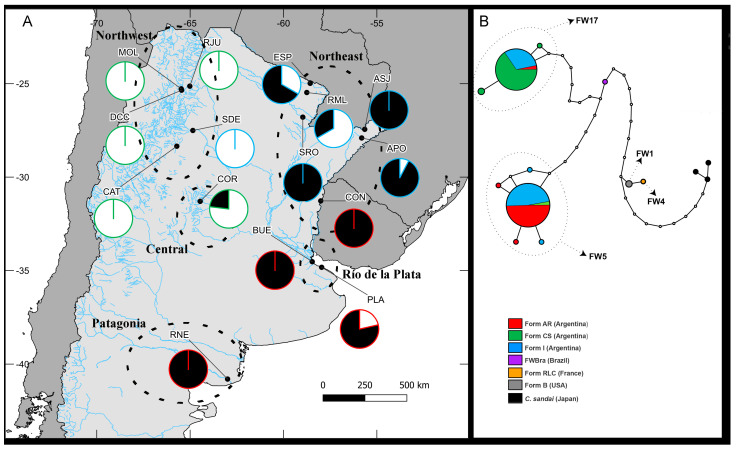
(**A**) Sampling locations and the geographic distribution of the COI haplotypes for *Corbicula* clams collected at 15 sites between 2015 and 2017. Pie charts depict the proportion of haplotypes observed at each site corresponding to the two freshwater lineages found within the country, with FW5 in black and FW17 in white. The border color of each pie chart indicates the identified form (see codes in panel (**B**)). Ellipses with dashed lines indicate distinct biogeographic zones covered in this study. (**B**). Unrooted statistical parsimony network of *Corbicula* clams based on 658 bp of COI, focusing on the 210 specimens from Argentina sequenced in this study and representative of the freshwater (FW) haplogroups FW5 and FW17. Solid lines represent single mutational steps, empty circles represent intermediate, unsampled haplotypes and circle sizes are proportional to the number of identical haplotypes. Haplotypes are color-coded based on the phenotypic form of each specimen (forms AR, CS, and I from Argentina were collected in this study, while the others come from the literature), and major FW haplogroups are indicated with dashed arrows and circles over the network. Codes and details regarding sampling sites can be found in Table 1.

**Figure 5 animals-14-01843-f005:**
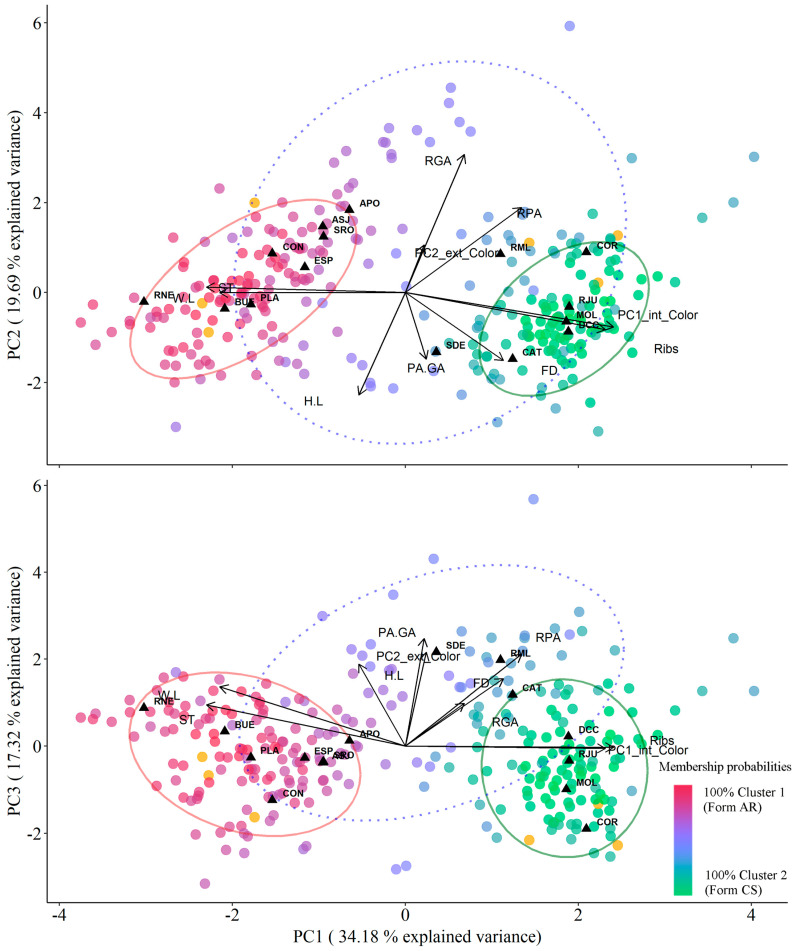
Principal component analysis of morphometric variables of 225 *Corbicula* clams sampled from 15 populations in Argentina between 2015 and 2017. Each data point represents an individual, with colors indicating the probability of belonging to different clusters based on the fuzzy cluster analysis. Orange data points indicate individuals with mismatches between COI lineage and morphotype. Population centroids are indicated by triangles. Ellipses represent the morphospace containing 90% of individuals from each cluster (solid lines) and intermediate individuals (dotted line) obtained from fuzzy cluster analyses. Variable codes correspond to those in Table 3, and population codes correspond to those in Table 1.

**Table 1 animals-14-01843-t001:** Identification of haplotypes of the mitochondrial cytochrome c oxidase subunit I (COI) gene in 15 *Corbicula* populations in Argentina, along with their correspondence with haplotypic groups and global forms, as defined in Pigneur et al. (2014) [14]. Intermediate forms are indicated as Form I.

ID	Region; Province	Water Body; Basin	Lat. Lon.	N	Haplotypic Group	Form
DCC	Northwest; Salta	Cabra Corral reservoir; Río de la Plata basin	−25.34; −65.45	15	FW17	CS
RJU	Northwest; Salta	Juramento River; Río de la Plata basin	−25.13; −65.01	11	FW17	CS
MOL	Northwest; Salta	Moldes irrigation ditch; Río de la Plata basin	−25.27; −65.47	15	FW17	CS
SDE	Northwest; Sgo. del Estero	Dulce river; Mar Chiquita basin	−27.50; −64.85	15	FW17	I
CAT	Northwest; Catamarca	Del Valle River; Del Valle basin	−28.34; −65.71	15	FW17	CS
COR	Center; Córdoba	Cosquín River; Mar Chiquita basin	−31.29; −64.46	13	FW17 (10); FW5 (3)	CS
PLA	Estuary; Buenos Aires	Río de la Plata estuary	−34.82; −57.95	14	FW5 (11); FW17 (3)	AR
BUE	Estuary; Buenos Aires	Río de la Plata estuary	−34.52; −58.45	14	FW5	AR
RNE	Patagonia; Río Negro	Negro River	−40.80; −62.99	15	FW5	AR
CON	Northeast; Entre Ríos	Ayuí Grande stream; Río de la Plata basin	−31.27; −58.00	15	FW5	AR
APO	Northeast; Misiones	Chirimay stream; Río de la Plata basin	−27.90; −55.81	14	FW5	I
ASJ	Northeast; Misiones	San Juan stream; Río de la Plata Basin	−27.44; −55.64	13	FW5 (12); FW17 (1)	I
SRO	Northeast; Chaco	de Oro River; Río de la Plata basin	−26.78; −58.96	14	FW5	I
ESP	Northeast; Formosa	Porteño River; Río de la Plata basin	−24.97; −58.56	12	FW5 (8); FW17 (4)	I
RML	Northeast; Formosa	Monte Lindo River; Río de la Plata basin	−25.47; −58.75	15	FW5 (5); FW17 (10)	I

**Table 2 animals-14-01843-t002:** Summary statistics for the two major haplogroups of *Corbicula* found in Argentina in this study. The table shows the sample size (n), nucleotide diversity (π), and number of specimens of each phenotypic form for each lineage, as well as the mean uncorrected genetic distance between the two haplogroups (p-distance, %).

	Haplogroup (Argentina)	
	FW5	FW17
n	111	99
*π*	0.00013	0.00012
Form AR	55	3
Form CS	3	66
Form I	53	30
p-distance	2.28%	

**Table 3 animals-14-01843-t003:** Results of the principal component analysis (PCA). The table shows the scores obtained for each independent variable for each factor. Values higher than 0.6 are indicated in bold font. PC1_int_Color and PC2_ext_Color: First two principal components scores obtained from the PCA conducted on values of color composition of the *Corbicula* shells; H:L: shell height to length ratio; W:L shell width to length ratio; ST: shell thickness; RGA: relative gill area; RPA: relative palp area, PA:GA: palp area to gill area ratio; FD: filament density.

Variable	Principal Component		
	1	2	3
PC1_int_Color	**0.89**	−0.21	−0.01
PC2_ext_Color	0.08	0.30	**0.65**
H:L	−0.20	**−0.64**	0.50
W:L	**−0.79**	0.00	0.36
ST	**−0.85**	0.03	0.25
RGA	0.25	**0.86**	0.26
Ribs	**0.86**	−0.23	−0.01
RPA	0.50	0.53	0.56
PA:GA	0.09	−0.41	0.57
FD	0.42	−0.42	0.41
Standard deviation	1.85	1.40	1.32
Proportion of variance	0.34	0.20	0.17
Cumulative proportion	0.34	0.54	0.71

## Data Availability

Original data will be published in open repositories (GenBank and Zenodo).

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
