# Peer review of "Morphological and Genetic Assessment of Invasive Corbicula Lineages in Southern South America: A Case Study in Argentina"

_animals, 2024, doi:10.3390/ani14131843_

Round 1

Reviewer 1 Report

Comments and Suggestions for Authors

Author Response

Thank you very much for taking the time to review this manuscript. Please find the detailed responses below

Comment 1: With so many factors, a flow chart might be helpful.

Thank you for the suggestion to include a flow chart. We appreciate your thoughtful consideration of our manuscript. However, upon careful review, we believe that the complexity of the factors involved may not lend themselves well to a flow chart in this instance. Instead, we have structured the information in a clear and concise manner within the text to ensure readability and understanding for our readers. We are confident that our approach effectively conveys the key processes and findings outlined in the study. Once again, we appreciate your valuable input and are open to further suggestions or feedback.

Comment 2: Another summary table also could be useful.

Another reviewer actually requested that we condense Tables 1 and 2 into a single table. However, we chose to retain both tables to provide more comprehensive information. Additionally, a supplementary file contains the entire raw dataset. Thus, we believe all relevant information that could be presented in tables is already included in the paper, and cannot find any further relevant information to add.

Comment 3: Some references were in Spanish (60, 61, 62).

This paper exhaustively treats Argentine populations of Corbicula spp. While we endeavoured to comprehensively cite all relevant information available in English, we have also included a few valuable sources for local populations published in Spanish and Portuguese. We have kept such reference to a minimum, and would be willing to remove them if the editor deems that absolutely necessary.

Regards,

Leandro Hünicken- On behalf of coauthors

Reviewer 2 Report

Comments and Suggestions for Authors

The manuscript Morphological and genetic assessment of invasive Corbicula lineages in southern South America: a case study in Argentina aims to review Corbicula's distribution in Argentina, discriminate extant lineages based on both morphological and genetic (COI) data, and describe variation in internal and external morphology using Argentine populations. 

After reviewing the manuscript, my recommendation is that the manuscript should be accepted after major revisions and another round of revisions. I endorse my suggestion, based on the points described below that deserve attention and need to be reviewed carefully:

Table I:

- I didn't understand how a population with FW5 and FW17 lineages only had the C/S or A/R form if you detected both lineages in sympatry. For example, in the COR population, there is the presence of both FW5 and FW17 lineages, but they only have C/S form. Maybe you didn't observe another form because the sample N was low? Say something about it. 

- what is Corbicula sp. form, it would be the intermediate morphotype I ? If yes, you need to change it here and along the manuscript, as well as in other Figures and Tables.

-combine this table with Table 2 (see my comment on table 2 next).

Figure 1: Add here a photo and description of those individuals with intermediate (I) forms found in your study and that were mentioned in the manuscript.

Figure 2: Add a size scale.

Line 198: Provide here the GenBank numbers.

Lines 298-302: The FWBra designates a COI lineage and not the morphotype of those individuals, which indeed exhibit an intermediate morphotype among A/R and C/S forms (Ludwig et al.2023). It should also be corrected in Table S2 because it does not represent the morphotype FWBra but the COI lineage (The identification column should be  Corbicula sp. FWBra and the Form column should be Intermediate morphotype or I).

Figure 4:

-Use the same acronyms as in Table 1+2 (new table), because there are some mismatches like PL (Fig. 4) and PLA (Table 1). Review all of them in the manuscript.

- Add ellipses or circles with the regions also evaluated together: the Río de la Plata estuary (PLA, BUE); Coastal (CON, APO, ASJ, SRO, ESP, RML); Central (COR); Northwest (DCC, MOL, RJU, SDE, CAT); and Patagonia (RNE).

-In Figure 4b, within the FW5 or FW17 haplogroups, other haplotypes appear at a lower frequency. Who are they? Why there is no indication of them in Table 1 or supplementary tables? If they are new COI haplotypes, please provide their COI sequences (maybe as supplementary information) and deposit them in the GenBank.

-indicate in the legend that forms A/R, C/S and I refer to the samples collected by your study and that the others come from the literature.

Table2: This table should be combined with Table 1 to demonstrate distribution by population. Here, you designate form I and in Table 1 no, please include form I appropriately in the new Table (1+2).

Line 347: What are these haplotypes? Is there any correspondence with those in Table S2? say something about

Lines 352-356: point out who these are in Fig.5

Table3: This table with mixed data makes no sense, as you show that there are morphological differences between the A/R, C/S, and I forms but, also, between lineages. Therefore, I suggest dividing by forms.

Figure5:

-Some acronyms' locations overlap and make identification differentiate them. I suggest moving the acronym's names to avoid their overlapping.

-identify those individuals (maybe with arrows) that stand out in the manuscript that could represent the hybrids and those that could represent cytonuclear mismatches (Buenos Aires and Cordoba).

Lines 389-392: This sentence needs to be complemented with the information that it is postulated that hybrids among Corbicula lineages could exhibit intermediate morphotypes, as those you identified. So,  is possible to identify intermediate morphotypes in some populations where FW5 and FW17 lineages are living in sympatry like PL, ESP, RML, COR, and APO.

Reviewer 3 Report

Comments and Suggestions for Authors

It has been a pleasure to read the manuscript. It is well-organized, well-written, and has an appropriate length. The authors use a combination of traditional (morphology) and modern (genetic) to define the extent and variability of Argentina's Corbicula lineages. As a consequence of this approach, the authors were able to match phenotypic variations with genetic lineages. Although the number of specimens used is on the low end, the study still provides a solid basis to understand further how these populations evolve/adapt to the changing environment at a large scale. 

Minor point

Although COI is spelled out in Table 1, spell it out in the text the first time it is mentioned.

Were the primers from the COI set designed for this study or from the literature?

Author Response

General comment: It has been a pleasure to read the manuscript. It is well-organized, well-written, and has an appropriate length. The authors use a combination of traditional (morphology) and modern (genetic) to define the extent and variability of Argentina's Corbicula lineages. As a consequence of this approach, the authors were able to match phenotypic variations with genetic lineages. Although the number of specimens used is on the low end, the study still provides a solid basis to understand further how these populations evolve/adapt to the changing environment at a large scale. 

Thanks! ?

Minor point

Comment 1: Although COI is spelled out in Table 1, spell it out in the text the first time it is mentioned.

Done

Comment 2: Were the primers from the COI set designed for this study or from the literature?

Response: Primers were from the literature. The correct citation was added (line 192, clean versión).

Regards,

Leandro Hünicken - On behalf of coauthors

Round 2

Reviewer 2 Report

Comments and Suggestions for Authors

The manuscript "Morphological and genetic assessment of invasive Corbicula lineages in southern South America: a case study in Argentina aims to review Corbicula's distribution in Argentina", discriminate extant lineages based on both morphological and genetic (COI) data, and describe variation in internal and external morphology using Argentine populations. 

I thank the authors for promptly responding to the requested suggestions and corrections, which significantly improved the quality of the results presented.